# Anxiety, Depression and Post-Traumatic Stress Disorder in Physicians Compared to Nurses during the COVID-19 Pandemic: An Observational, Cross Sectional, Multicentric Study

Roberto Lupo [1], Stefano Botti [2], Alessandra Rizzo [3], Alessia Lezzi [4], Antonino Calabrò [5], Luana Conte [6,7], Cosimo Petrelli [1], Cosimo Longo [1] and Elsa Vitale [8,*]

[1] San Giuseppe da Copertino Hospital, Local Health Authority Lecce, 73100 Lecce, Italy; roberto.lupo@uniba.it (R.L.); cosimodrpetrelli@libero.it (C.P.); cosimo.longo@asl.lecce.it (C.L.)
[2] Haematology Unit, Azienda USL-IRCCS Reggio Emilia, 42122 Reggio Emilia, Italy; botti@ausl.re.it
[3] Local Health Authority Taranto, 74121 Taranto, Italy; alessandra.rizzo95@libero.it
[4] National Cancer Association Lecce, 73100 Leccel, Italy; alessia.lezzi@gmail.com
[5] Nuovo Ospedale degli Infermi Hospital, Local Health Authority Biella, 13900 Biella, Italy; anto.cala76@gmail.com
[6] Laboratory of Biomedical Physics and Environment, Department of Mathematics and Physics "E. De Giorgi", University of Salento, 73100 Lecce, Italy; luana.conte@unisalento.it
[7] Laboratory of Interdisciplinary Research Applied to Medicine (DReAM), University of Salento and ASL (Local Health Authority), 73100 Lecce, Italy
[8] Mental Health Centre, Local Health Authority Bari, 70121 Bari, Italy
* Correspondence: vitaleelsa@libero.it

**Abstract:** (1) Background: The coronavirus pandemic has highlighted the precarious health situation of our country, thanks to the grueling workloads caused by understaffing and fear of contracting COVID-19. By considering this critical situation, frontline healthcare professionals who have been directly involved in the diagnosis, treatment, and care of SARS-CoV-2 patients are now at risk of developing psychological distress and other mental health symptoms, accomplices of the fear of contracting the COVID-19 and the exhausting workloads. (2) Methods: An observational, cross-sectional, multicenter study was conducted by administering an online questionnaire to all Italian physicians and nurses who worked during the COVID-19 pandemic. The questionnaire consists of socio-demographic characteristics, an assessment of anxiety levels with the State-Trait Anxiety Inventory (STAI), which also assessed trait and state anxiety, the Beck Depression Inventory (BDI) to evaluate the condition of depressive severity, and, finally, the Impact of Event Scale–Revised (IES–R), which was administered in order to quantify the post-traumatic stress disorder (PTSD) among the participants. (3) Results: A total of 770 Italian healthcare workers were enrolled in this study. Of these, 95 (12.30%) were physicians and 675 (87.70%) were nurses. By considering PTSD, anxiety, and depression levels between the physicians and nurses recruited, a significant difference was reported in the STAI-1 assessment, as both physicians and nurses reported slight and moderate levels (*p* = 0.033). (4) Conclusions: Physicians and nurses, who have been subjected to physical impoverishment, with the infinite physical forces spent to support the pace of work at the limits of the possible, but above all mental capacity, with the anxiety of having to face an unknown enemy, such as COVID-19. This has resulted in a significant increase in anxiety, depression, post-traumatic symptoms, and sleep disturbances, with possible repercussions not only on the quality of life of the physicians and nurses but also on the quality of assistance provided.

**Keywords:** anxiety; COVID-19; depression; nurse; post-traumatic stress disorder; physician

## 1. Introduction

The pandemic that we have been witnessing for two years has created a real danger not only for the specific symptoms and complications of the SARS-CoV-2 syndrome but also, above all, for the global health of people, which has been under constant threat [1]. In fact, it has seriously undermined the psychic and resilience structure of health professionals, causing long-term domino effects and leading to the insinuation of mental dynamics harmful to the workers, such as anxiety, depression, and post-traumatic stress disorders. It must be considered that the coronavirus emergency has added further complexity to the working world, significantly changing the way in which operators have conducted and organized their work and their lives. It has been difficult to understand and frame the changes that the state of emergency brought to everyday life. This resulted in a period of isolation that led the population to temporarily freeze their lives. Studies on the outcomes of the SARS-CoV-2 virus have highlighted both organic compromise and the possible onset of neurological and/or psychiatric pathologies that can persist in subsequent years if not adequately treated [2]. Due to the COVID-19 emergency, health professionals have been engaged on the front lines, dealing with critical conditions that require greater expertise and experience in various care settings, despite being constantly exposed to the risk of infection and emotional overload.

The ever-increasing number of confirmed and suspected cases, the overwhelming workload, the exhaustion of wearing personal protective equipment, the widespread media coverage, the lack of specific drugs, and the feeling of being inadequately supported have characterized the experiential experience of many healthcare workers [3]. Healthcare workers have faced enormous pressures, including a high risk of infection and inadequate protection from contamination, overwork, frustration, discrimination, isolation, patients with negative emotions, a lack of contact with their families, and fatigue, generating post-traumatic stress disorder [4]. The serious situation is now causing mental health problems such as stress, anxiety, depressive symptoms, insomnia, denial, anger, and fear. The protection of the mental health of these health workers is, therefore, important for the control of the epidemic and their long-term health [5]. From a study conducted on a sample of 1257 healthcare workers in 34 hospitals, high levels of depression, anxiety, insomnia, and distress emerged in China [3].

Another study involving 1563 participants found that over a third of medical staff experienced symptoms of insomnia during the COVID-19 epidemic [6].

As shown by the literature, some factors related to the pandemic, such as the danger of the disease and the restrictive measures adopted, have been a source of concern and anxiety among the general population [7,8] and among healthcare professionals (HCPs), leading to an increased risk of developing psychiatric symptoms [8,9]. Nurses were more prone to developing burnout and stress disorders during the pandemic outbreak [3,10], due to various factors such as their closeness to patients, higher work rate, increasing emotional demands, and the concern of being infected with COVID-19 and passing it on to others [11].

The mental health of nurses themselves was also investigated by a study conducted in Italy, in the hematological field, where moderate levels of insomnia and stress emerged [12]. However, to date, the studies conducted on the current psycho-physical conditions of health workers are limited.

The aim of the present study was to analyze any differences that existed between Italian physicians and nurses in anxiety, depression, and PTSD levels during the COVID-19 pandemic.

## 2. Materials and Methods

### 2.1. Study Design

An observational, cross-sectional, multicentric study was carried out from March 2020 to June 2020 by administering an online questionnaire to all Italian physicians and nurses who were employed during the COVID-19 pandemic.

## 2.2. Questionnaire

In the first part of the questionnaire, socio-demographic characteristics were collected among participants. Specifically:

- professional role, such as physician or registered nurse;
- sex, female or male;
- years of work experience, if the respondent worked more or less than 1 year;
- ward assigned, if the ward treated COVID-19 patients or not;
- their own health condition perceptions, classified into excellent, acceptable, or bad.

In the second part of the questionnaire, anxiety levels were assessed thanks to the State-Trait Anxiety Inventory (STAI), which evaluated both trait and state anxiety [13], as the literature has shown a good reliability and validity of this instrument [14–16]. The first 20 items (STAI-1) evaluated trait anxiety, by highlighting a tense and worried condition. The other 20 items (STAI-2) assessed the state-anxiety dimension, which underlined the state-anxiety condition and the capability to feel calm and secure. For each item, a 4-point Linkert scale was associated, which varied from "1" meaning "almost never" to "4" meaning "almost always". By summing the first 20 items, for the trait-anxiety dimension, and by summing the other 20 items, for the state-anxiety dimension, scoring ranges were obtained. Specifically, for values between 20–39, the absence of anxiety was indicated; for values between 40–50, a slight anxiety disorder was identified; for scores between 51–60, a moderate anxiety level was highlighted; and, finally, for values between 61–80, a severe anxiety condition was identified.

Then, the Beck Depression Inventory (BDI) [17] in its brief form was included, in order to assess depressive severity condition. This version of the inventory consisted of 13 items, in which a Linkert scale was associated which varied from 0 to 3. By summing all the answers, a value was obtained which could be referred to specific validated ranges, namely for values from 0 to 4, none or a minimal depression condition was assessed. For values between 5 and 7, a mild depression condition was evaluated; for values between 8 and 15, a moderate depressive condition was identified; and, finally, for values more than 16, a severe condition was assessed.

Finally, included in the questionnaire was the Impact of Event Scale–Revised (IES–R) [18], in order to quantify post-traumatic stress disorder (PTSD) among participants. The IES–R contained a total of 22 items including a list of problems that respondents sometimes could perceive following events stressors of life. A 4-point Likert rate was associated, which varied from 0 meaning "not at all", to "4" meaning "extremely". Additionally, three subdimensions of PTSD were also evaluated, such as:

- "Avoidance", exploring how the subject avoided thinking about the traumatic happening;
- "Intrusiveness", defining how the subject could not help but think about the stressful event;
- "Hyperarousal", evaluating how much anger and irritability the interviewee felt indefinitely.

The maximum mean score of each of the 3 subscales was 4, hence the maximum total mean score of the IES–R scale was 12. Lower scores were better, and an IES–R total score of 33 or higher out of a maximum score of 88 meant the probable presence of post-traumatic stress disorder (PTSD).

$\alpha$-Cronbach of the questionnaire created was assessed at 0.760, by highlighting a good internal consistence of each item. Specifically, $\alpha$-Cronbach for the STAI-1 was assessed at $\alpha = 0.666$; for the STAI-2, $\alpha = 0.760$; for the BDI, $\alpha = 0.697$; and for the IES–R, $\alpha = 0.705$.

## 2.3. Data Analysis

Data were collected in an Excel data sheet and then processed through the use of the Statistical Package for Social Sciences (SPSS), version 20. Socio-demographic data were assessed as frequencies and percentages for each professional subgroup and *chi-*

*squared* test was performed to assess any differences. Additionally, the State-Trait Anxiety Inventory (STAI-1 and the STAI-2) for anxiety and the Beck Depression Inventory (BDI) for depression disorder were evaluated as categorical variables and assessed as frequencies and percentages. Meanwhile, the total IES–R values and their related subdimensions' values were presented as means and standard deviations. Chi-squared test was assessed between the physician the nursing group for anxiety and depression levels, while the t-test for independent sample was performed to evaluate sampling differences between the physician group and the nursing group for the IES–R values and its related subdimensions too. All values < 0.05 were considered as statistically significant.

### 2.4. Ethical Considerations

The present study was carried out according to the World Medical Association's Declaration of Helsinki [19]. All Italian physicians and nurses who were employed in an Italian healthcare setting during the COVID-19 pandemic and voluntarily agreed to participate to the study were enrolled.

In the first part of the questionnaire, a presentation with the ethical characteristics of the study was presented. The questionnaire was spread through some Facebook and Instagram pages, in order to reach a higher number of participants during the COVID-19 pandemic. No consent was requested from the Ethics Committee, but it was underlined that participation was voluntary and only those who were interested in participating were given an informed consent form, which reminded them of the voluntary nature of participation.

## 3. Results

A total of 770 Italian healthcare workers were enrolled in this study. Of these, 95 (12.30%) were physicians and 675 (87.70%) were nurses. All sampling characteristics were reported in Table 1. Significant differences were assessed only in the gender composition of the sample collected, since females were more numerous than males both for physicians and nurses ($p < 0.001$).

**Table 1.** Sampling characteristics ($n = 770$).

| Characteristic | Physician 95 (12.30%) | Registered Nurse 675 (87.70%) | *p*-Value |
|---|---|---|---|
| Sex | | | |
| Female | 43/95 (45.26%) | 529/675 (78.37%) | >0.001 * |
| Male | 52/95 (54.74%) | 146/675 (21.63%) | |
| Work experience | | | |
| >1 year | 23/95 (24.21%) | 216/675 (32.00%) | 0.155 |
| <1 year | 72/95 (75.79%) | 459/675 (68.00%) | |
| Ward assigned | | | |
| COVID-19 | 16/95 (16.84%) | 142/675 (21.04%) | 0.416 |
| Other | 79/95 (83.16%) | 533/675 (78.96%) | |
| Health perception | | | |
| Excellent | 31/95 (32.63%) | 164/675 (24.30%) | |
| Acceptable | 61/95 (64.21%) | 485/675 (71.85%) | 0.215 |
| Bad | 3/95 (3.16%) | 26/675 (3.85%) | |

* $p < 0.05$: statistically significant.

By considering PTSD, anxiety, and depression levels between physicians and nurses recruited, a significant difference was reported in the STAI-1 assessment, as both physicians and nurses reported slight and moderate levels ($p = 0.033$). Meanwhile, in the STAI-2 assessment for depression and in the PTSD assessment, no statistical significances were reported. However, low values in the PTSD were reported, and 4.90% of nurses and 1% of physicians reported a severe depression condition too (Table 2). All these findings

confirmed the health condition considered by each participant, since most physicians (7.90%) and nurses (63.00%) considered their health status as "acceptable" (Table 1).

**Table 2.** Anxiety, depression, and PTSD levels among physicians and nurses.

| Anxiety, Depression, and PTSD Levels | Physician 95 (12.30%) | Registered Nurse 675 (87.70%) | *p*-Value |
|---|---|---|---|
| STAI-1 [range value] | | | |
| Absence [20–39] | 0 (0%) | 0 (0%) | |
| Slight [40–50] | 47 (6.10%) | 262 (34%) | |
| Moderate [51–60] | 47 (6.10%) | 412 (53.50%) | 0.033 [a]* |
| Severe [61–80] | 1 (0.10%) | 1 (0.10%) | |
| STAI-2 [range value] | | | |
| Absence [20–39] | 0 (0%) | 2 (0.30%) | |
| Slight [40–50] | 50 (6.50%) | 282 (36.60%) | |
| Moderate [51–60] | 44 (5.70%) | 350 (45.50%) | 0.075 [a] |
| Severe [61–80] | 1 (0.10%) | 41 (5.30%) | |
| BDI [range value] | | | |
| None or minimal [0–4] | 56 (7.30%) | 344 (44.70%) | |
| Mild [5–7] | 15 (1.90%) | 151 (19.60%) | |
| Moderate [8–15] | 16 (2.10%) | 142 (4.90%) | 0.213 [a] |
| Severe [<16] | 8 (1.00%) | 38 (4.90%) | |
| IES–R total | 2.99 ± 2.89 | 3.54 ± 2.77 | 0.444 [b] |
| Avoidance | 0.99 ± 0.92 | 1.10 ± 0.86 | 0.343 [b] |
| Intrusiveness | 0.99 ± 0.96 | 1.18 ± 0.95 | 0.637 [b] |
| Hyperarousal | 1.02 ± 1.00 | 1.19 ± 0.94 | 0.277 [b] |

* $p < 0.05$: statistically significant; [a] *chi-squared* test; [b] *t*-test for independent sample.

## 4. Discussion

The aim of this research study was to analyze the differences between Italian physicians and nurses, by considering the possible psycho-physical sequelae, such as anxiety, depression, and post-traumatic stress disorder during the COVID-19 pandemic. To date we were faced with an epidemiological picture that saw a dramatic death toll from COVID-19 among the 135 million health and care workers in the world (healthcare workers or HCWs). According to a new WHO report, at least 115,500 healthcare workers, between January 2020 and May 2021 lost their lives due to the pandemic.

Deaths among health workers were about five times higher than the average for the general population [20]. Although there were not yet sufficiently large studies on the topic of Long COVID-19 Syndrome, so it was possible to state that the range of healthcare workers affected from the medium- to long-term ranged between 10% and 20%, which was, in absolute numbers, between 13,000 and 20,000 healthcare workers involved [20]. One of the heaviest consequences of the spread of COVID-19 was the current health crisis that attacked national health systems around the world. Professionals in each sector, with their different roles and tasks, were called to face an emergency of enormous magnitude, which affected not only workloads and physical fatigue but also their psychological health.

Healthcare professionals currently constitute a high-risk group in developing a wide range of physical/mental problems following direct or indirect work with COVID-19 patients and were particularly exposed to the threat of transmission due to their frontline work with patients with high viral loads and with individual protective devices that were not always optimal. In a study conducted among the general population of Hong Kong and health workers in Taiwan, it was found that participants in Hong Kong had less concern about the amount of personal protective equipment (PPE) (3.6%) than healthcare workers (97.4%). Significantly lower psychological distress (mean (SD) = 0.16 (0.39) in the Hong Kong general population versus 0.47 (0.59) in Taiwanese health workers) [19] also emerged in the study by Sagaon-Teyssier et al. [21], in which the availability of PPE reduced the risk of developing depression (51%), insomnia (43%), and anxiety (49%).

In our study, the majority of participants were nurses (87.70%). Of these, 68.70% were female, and 63% had been employed less than one year. As far as physicians were concerned, they were 12.30% of the sample. Of these, 6.80% were male and 7.90% perceived their levels of health as acceptable. A literature review showed a high overall psychological impact of the COVID-19 pandemic among healthcare professionals in the general population and patients with pre-existing diseases or COVID-19. The most common indicators of psychological impact, reported in the studies considered, were anxiety and depression, with respective prevalence of 33% (28–38%) and 28% (23–32%). Patients with pre-existing conditions or COVID-19 had a significantly higher prevalence of anxiety and depression than healthcare professionals and the general population [22].

A review on the psychological impact of COVID-19 on healthcare workers in African countries showed rates of 9.5% to 73.3% for anxiety disorders and rates of 12.5% to 71.9% for depression [23]. Furthermore, several systematic reviews [24,25] explained sleep disorders in healthcare workers during the COVID-19 pandemic. For example, a systematic review of 168 revealed a majority with sleep disturbance during the pandemic, with a strong correlation with anxiety, in 57% of COVID-19 patients and 31% of healthcare professionals, which was only 18% in the general population [25].

By considering PTSD, anxiety, and depression levels between the physicians and nurses recruited, a significant difference was reported in the STAI-1 assessment, as both physicians and nurses reported slight (physician: 6.10%; registered nurse: 34%) and moderate levels (physician: 6.10%; registered nurse: 53.50%) ($p = 0.033$). From a systematic review and meta-analysis carried out in 2021 by Li et al. [23], with the aim of providing updated estimates of the prevalence of depression, anxiety, and post-traumatic stress disorder (PTSD) among healthcare professionals during the COVID 19 pandemic, in 57 studies the aggregate prevalence of moderate anxiety was 22.1%, where individual study estimates ranged from 5.2% to 89.7%, while the prevalence of mild anxiety was 38.3%. Of these, only 27 studies provided data regarding the proportion of participants in close contact with COVID-19 patients, in which there was a higher prevalence of anxiety than in studies in which participants in close contact were less than 50% [23]. It is clear that nursing work is more conspicuously affected by anxiety disorder and related aspects. In fact, as reported in the study by Cai et al. [10], the nurses were the most affected by nervousness and anxiety when they were inside the ward compared to the medical one; in contrast to past epidemics, in which 85% of doctors reported higher levels of pressure in the workplace and did not have enough resources to cope with it. Furthermore, from a review by Li et al. [23], it emerged that in 55 studies, the aggregate prevalence of depression was 21.7%, with individual study estimated as ranging from 5.3% to 57.6%, while mild depression was 36.1%. The variation of these estimates also depends on the regions compared: the estimates were highest in the Middle East, while the lowest in North America and East Asia [23]. The last section of the investigation tool detected the possible presence of post-traumatic stress disorder thanks to the IES–R total, a scale that evaluates the impact of stress on the individual after traumatic events. Study results indicate that nurses reported higher rates of PTSD than physicians, in line with another Italian study. Nowhere did it emerge that pandemic knowledge and the nursing role influenced depressive conditions ($p = 0.006$), as nurses recorded more normal scores (52.5%) than the general population (19.5%). On the other hand, data recorded no statistical significance between nurses and the general population for concern anxiety disorders ($p = 0.265$). Additionally, significant correlations were found between knowledge and anxiety levels ($p = 0.024$) and nursing and the general population role and anxiety levels ($p = 0.005$) too [26]. In fact, due to staff shortages, nurses face a greater physical and mental load, as well as greater hours of assistance in close contact with patients, than doctors; in fact, it emerged in a study that greater contact with more serious patients is linked to higher IES/Impact of Event Scale–Revised scores [27]. In an Italian study involving a sample of 1500 healthcare workers, 93% developed psychosomatic disorders including nausea, nightmares, and palpitations [28]. In nine studies, the combined estimate of the prevalence of moderate PTSD was 21.5%, with a per-study variation ranging from 2.9% to 49.5% [29].

A recent review focused attention on how previous epidemics also had a psychological impact that lasted years. Forty-four studies were included in the review. Between 11% and 73.4% of healthcare workers, including mainly physicians, nurses, and support staff, reported symptoms of post-traumatic stress that persisted after 1–3 years, by 10–40%. Depressive symptoms were reported by 27.5–50.7%, insomnia in 34–36.1% and severe anxiety by 45%. General psychiatric symptoms during epidemics ranged from 17.3% to 75.3%; high levels of work-related stress are reported by between 18.1% and 80.1% [28]. Events of this magnitude had a double impact on the mental health of the population, both in the immediate and in the long term. Meta-analyses that investigated the impact of COVID-19 on mental health found a significant increase in anxiety, depression, post-traumatic symptoms, and sleep disturbances. Salari and colleagues [30] analyzed the results of 17 studies and reported overall that while in Asia there was a greater prevalence of anxiety symptoms, in Europe the depressive and post-traumatic reactions seemed to be prevalent.

In Italy, Castelli and colleagues [31] found rates of 20% for PTSD, 69% for anxiety disorder, and 31% for depressive symptoms. The presence of interpersonal conflicts, frequent use of social media, reduced resilience skills, and poor social support was also reported [32]. In Italy, one of the few support initiatives for healthcare workers involved at the forefront of COVID-19 patient care was promoted by the National Institute for Insurance against Accidents at Work (Inail) which activated, in collaboration with the National Council of Order of Psychologists (Cnop), a national initiative aimed at promoting psychological support services addressed to healthcare workers. This initiative involved the creation of the link of a task force of psychologists in health facilities, in order to implement continuous monitoring. Further initiatives were subsequently implemented by some hospitals such as the Local Health Authority of Chieti, thanks to the use of "errare humanum est" that represented the relationship as a clinical risk management tool to detect early signs of psychological deviations of the operators involved in the COVID-19 emergency, in order to prevent discomfort conditions at several levels and ensure high standards of safety of care. The Tor Vergata Polyclinic in Rome, Italy, also promoted an emotional defusing as an early intervention applicable in the hospital. A further initiative was implemented by the "Città Della Salute e Della Scienza" hospital in Turin, through a COVID-19 social call center, in connection with third sector associations, the Red Cross and Civil Protection, to support the employees operating in several COVID-19 departments [33]. Abasi [34] argues how fundamental it is to cultivate resilience and mitigate stress through practices of self-awareness and "compassion" of one's moods, attribution of value to work, healthy lifestyle behaviors, harmonization of service with family expectations, and connection with colleagues, relatives, and friends as well as using telematic tools [34]. Bulut et al.'s studies [35,36] show that a higher PTSD rate and symptoms of insomnia among nurses demonstrate that the nursing profession needs to receive more attention in terms of psychological support. Possible burnout for operators involved in the health emergency must be prevented, by reducing the risk of developing PTSD for frontline workers and by reducing the risk of developing "corridor syndromes", in which employees can "experience the transition from the workplace to the private one and vice versa without interruption".

In light of the results that emerged from our study and from those highlighted by other studies, it is clear that there is a need for work organizations to support health professionals by establishing support services within each individual care center [37,38].

*Limitations*

The present study had several limitations. First of all, the research method, since the questionnaire was spread in an online mode though some "spontaneous" Facebook and Instagram pages, created bias concerning the enrollment number of healthcare professionals involved, without any sampling-size assessment. Therefore, the nurses and physicians recruited were merely random, and the collected number between participants was not balanced. Additionally, the participants who worked in COVID-19 departments and who

joined the survey were present in a limited manner. The questionnaire was administered online, which could influence the recruitment of participants who enrolled.

## 5. Conclusions

The results of our study draw attention to the importance of prevention strategies and, above all, indicate the need to explore the need for constant psychological support, especially for nurses.

The impact of COVID-19, on the mental health of physicians and nurses, caused a significant increase in anxiety, depression, post-traumatic symptoms, and sleep disturbances. In the long run, these symptoms can damage the psycho-physical health of the healthcare professional and the health of the patient with a reduced quality of care.

The attitude with which one faces a stressful condition and resilience, both as an individual and as a community, are ways of responding that can fortunately be learned, improved, and strengthened. We need to focus on new aspects such as a balanced coexistence with the virus [28]. In the light of the results obtained, it would also be useful to investigate suicidal ideation and the predictors of intention to leave the profession among physicians and nurses, to be considered as emerging problems in the healthcare world.

**Author Contributions:** Conceptualization, E.V. and R.L.; methodology, E.V.; software, E.V.; validation, E.V.; formal analysis, E.V.; investigation, A.L., R.L., A.C. and A.R.; resources, A.L. and A.R.; data curation, S.B.; writing—original draft preparation, R.L.; writing—review and editing, C.P., C.L., R.L. and L.C. All authors have read and agreed to the published version of the manuscript.

**Funding:** This research received no external funding.

**Institutional Review Board Statement:** Ethical review and approval were waived for this study due to the nature of the study: dealing with an online, observational study. Both Italian nurses and nursing students, who voluntarily agreed to participate in the study, were enrolled. All the ethical concerns of the study were stated in the first part of the questionnaire, in agreement with the principles of the Italian data protection authority (DPA).

**Informed Consent Statement:** Informed consent was obtained from all subjects involved in the study. Written informed consent for publication was obtained from participating nurses.

**Data Availability Statement:** Data are available from the corresponding author.

**Acknowledgments:** The authors acknowledged all participants who voluntary agreed to participate in the present study.

**Conflicts of Interest:** The authors declare no conflict of interest.

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
