# Peer review of "Anxiety, Depression and Post-Traumatic Stress Disorder in Physicians Compared to Nurses during the COVID-19 Pandemic: An Observational, Cross Sectional, Multicentric Study"

_psych, doi:10.3390/psych4030036_

Round 1

Reviewer 1 Report

Title - please add research design to the title

Intruduction - well written 

Methodology - you calculated Crombach alpha for all instruments used in second part together. Please show Crombach alpha for each instruments. 

Ethical consideration - suggest to state that you followed Helsinki declaration and cite it in references. 

Results and discussion well written. 

Author Response

Rebuttal letter

Anxiety, Depression and Posttraumatic Stress Disorder in physicians compared to nurses during the Covid-19 pandemic: an observational, cross sectional, multicentric study

We thank the Referee for the valuables comments.

Ref. No. 1

R1. Title - please add research design to the title

A1. The study design was added in the title

R2. Intruduction - well written 

A2. The Introduction section was the same.

R3. Methodology - you calculated Crombach alpha for all instruments used in second part together. Please show Crombach alpha for each instruments. 

A3. Cronbach alpha values were assessed.

R4. Ethical consideration - suggest to state that you followed Helsinki declaration and cite it in references. 

A4. The Reviewer’s suggestion was added in the “Ethical consideration” sub section.

R5. Results and discussion well written.

A5. Results and discussion were the same versions.

Changes and modifications have been marked in red color in the text to speed up reviewing.

I feel that the revised version of our manuscript, now, is ready for acceptance.

Sincerely yours,

Elsa Vitale

Bari, 30.06.2022

Reviewer 2 Report

The study surveyed a wide range of RNs and Physicians in Italy to estimate Anxiety, Depression, and Posttraumatic Stress Disorder. The study has exciting results with enough samples. The introduction, methods, and discussion are well developed however, there is more need to develop in the results section; most of my comments are for the result section:

Major comments:

1) T2, compared the categorical variables for four different scales, please add the scale values for all measures.

2) I would develop a boxplot to compare the four measures between PHYs and RNs by grouping them by gender.

3) Table 1. report % for PHY and RN, separately, for example, for PHYs: Female=43/95=xx% and RN+529/675, the same for other values, then run a chi-sq to see the sig diff.

4) Table 2. report % for PHY and RN, separately

Minor comments:

1) Add citation: line 71: a study conducted on a sample of 1,257 healthcare workers in 34 hospitals, high levels of depression, anxiety, insomnia, and distress emerged in China.

Add citation: line 204: Deaths among health workers are about five times higher than the average for the general population.

Add citation: between 13,000 and the 20,000 health workers involved in Italy?

Add citation: 267, in line with another Italian study.

Introduction: Please a few studies from OCED countries.

Discussion: I would add a few lines to discuss the recommendations for RNs and PHYs and the type of screening or monitoring that may help PHYs and RNs. I would also add a few comments on treating or providing services to PHYs and RNs facing depression, etc.

Edit: The MS need to be edited slightly.

Author Response

Rebuttal letter

Anxiety, Depression and Posttraumatic Stress Disorder in physicians compared to nurses during the Covid-19 pandemic: an observational, cross sectional, multicentric study

Ref. No. 2

We thank the Referee for the valuables comments.

The study surveyed a wide range of RNs and Physicians in Italy to estimate Anxiety, Depression, and Posttraumatic Stress Disorder. The study has exciting results with enough samples. The introduction, methods, and discussion are well developed however, there is more need to develop in the results section; most of my comments are for the result section:

Major comments:

R1. T2, compared the categorical variables for four different scales, please add the scale values for all measures.

A1. The scale values were added for all the scales considered.

R2. I would develop a boxplot to compare the four measures between PHYs and RNs by grouping them by gender.

A2. Since the manuscript focused on comparison between physicians and nurses on their psychological health, the Authors considered the boxplot a further tool which could not focus on the data presented.

R3. Table 1. report % for PHY and RN, separately, for example, for PHYs: Female=43/95=xx% and RN+529/675, the same for other values, then run a chi-sq to see the sig diff.

Table 2. report % for PHY and RN, separately.

A3. Both Table 1 and table 2 were organized according to the Reviewer’s suggestion.

R4. Add citation: line 71: a study conducted on a sample of 1,257 healthcare workers in 34 hospitals, high levels of depression, anxiety, insomnia, and distress emerged in China.

Add citation: line 204: Deaths among health workers are about five times higher than the average for the general population.

Add citation: between 13,000 and the 20,000 health workers involved in Italy?

Add citation: 267, in line with another Italian study.

 A4. Citations were added.

R5. Introduction: Please a few studies from OCED countries.

A5. The suggestion was considered.

R6. Discussion: I would add a few lines to discuss the recommendations for RNs and PHYs and the type of screening or monitoring that may help PHYs and RNs. I would also add a few comments on treating or providing services to PHYs and RNs facing depression, etc.

A6. The discussion section was improved.

Changes and modifications have been marked in green color in the text to speed up reviewing.

I feel that the revised version of our manuscript, now, is ready for acceptance.

Sincerely yours,

Elsa Vitale

Bari, 30.06.2022

Reviewer 3 Report

 The manuscript has compared the psychological effects of COVID-19 on physicians and nurses working during COVID pandemic. The results are supported by the data. However, the results are not properly discussed in the discussion section. Please discuss the results of this study. Additionally, the initial three paragraphs of the discussion section are poorly cited, please cite. There are some typos, please check.

In the limitation of the study section, please discuss if the data was analyzed after matching to various variables or confounding factors like age, BMI, previous history of psychological issues, time/duration of service in the hospital, place of working like in ICU or inpatient ward or OPDs.  etc. Please also include inclusion and exclusion criteria of the study, although the authors mentioned that inclusion was random. In that case please describe the confounding factors of this study.

Author Response

Rebuttal letter

Anxiety, Depression and Posttraumatic Stress Disorder in physicians compared to nurses during the Covid-19 pandemic: an observational, cross sectional, multicentric study

We thank the Referee for the valuables comments.

Ref. No. 3

R1.  The manuscript has compared the psychological effects of COVID-19 on physicians and nurses working during COVID pandemic. The results are supported by the data. However, the results are not properly discussed in the discussion section. Please discuss the results of this study. A1. The results are better discussed in the discussion section.

R2. Additionally, the initial three paragraphs of the discussion section are poorly cited, please cite. A2. Citations were added.

R3. There are some typos, please check.

A3. Typos were corrected.

R4. In the limitation of the study section, please discuss if the data was analyzed after matching to various variables or confounding factors like age, BMI, previous history of psychological issues, time/duration of service in the hospital, place of working like in ICU or inpatient ward or OPDs.  etc. Please also include inclusion and exclusion criteria of the study, although the authors mentioned that inclusion was random. In that case please describe the confounding factors of this study.

A4. The limitation section was also improved.

Changes and modifications have been marked in blue color in the text to speed up reviewing.

I feel that the revised version of our manuscript, now, is ready for acceptance.

Sincerely yours,

Elsa Vitale

Bari, 30.06.2022

This manuscript is a resubmission of an earlier submission. The following is a list of the peer review reports and author responses from that submission.

Round 1

Reviewer 1 Report

Dear authors,

congratulations to this manuscript, the topic is novel and interesting. I am sure that this article will raise great interest of the international audience. However, I have few suggestions for improvement. I suggest to add research design to the title. Try to not use so many abbreviations in the abstract. 

Introduction is well written. Methodology sufficiently described. Do you have Cronbach alpha for the instruments used in the study?

Results are nice presented. Discussion is sufficient . Conclusion good.

Author Response

vcx

Rebuttal letter

Anxiety, Depression and Posttraumatic Stress Disorder in

Physicians compared to nurses during the Covid-19 pandemic

We thank the Referee for the valuables comments.

Ref. No. 1

Q1. Dear authors,

congratulations to this manuscript, the topic is novel and interesting. I am sure that this article will raise great interest of the international audience. However, I have few suggestions for improvement. I suggest to add research design to the title. Try to not use so many abbreviations in the abstract. 

Introduction is well written. Methodology sufficiently described. Do you have Cronbach alpha for the instruments used in the study?

Results are nice presented. Discussion is sufficient . Conclusion good.

R1. Thank you for your comfortable revisions. Cronbach alpha for the instruments adopted was also assessed.

Changes and modifications have been marked in red color in the text to speed up reviewing.

I feel that the revised version of our manuscript, now, is ready for acceptance.

Sincerely yours,

Elsa Vitale

Bari, 13.05.2022

Reviewer 2 Report

The submission has the following significant flaws.

1. The authors claimed that they recruited 700 participants; however, 95 physicians plus 675 nurses exceed 700.

2. The entire manuscript is hard to follow as the authors use long paragraphs throughout the manuscript (e.g., only one paragraph in Introduction) 

3. The authors did not control important confounders for the physician-nurse comparisons. They only used independent t-tests to examine the differences in the outcome measures. 

4. There are many unclear terms (e.g., STAY-1 and STAY-2; BDI). I know that the authors used Beck Depression Inventory; however, they did not introduce it. Instead, they said "Depression Inventory". Also, it is weird that they abbreviated State-Trait Anxiety Inventory into STAY instead of STAI. Moreover, they did not define what is STAY-1 and STAY-2.

5. There are some unclear sentences. For example, "The relationship between family caregiving and the mental health of emerging young adult caregivers [14–16]."

6. The authors mentioned "An observational, cross sectional, multicentric study, was carried out from March 2017 to December 2019 by administering an on line questionnaire in all Italian physicians and nurses who were employed during the Covid-19 pandemic." How can this happened? The pandemic begun from end of 2019!!

7. There are quite a lot of typos (e.g., In the first art of the questionnaire; this should be In the first part of the questionnaire).

8. Quite a lot of important references were not cited. Below are some examples.

Patel BR, Khanpara BG, Mehta PI, Patel KD, Marvania NP. Evaluation of perceived social stigma and burnout, among health-care workers working in covid-19 designated hospital of India: A cross-sectional study. Asian J Soc Health Behav 2021;4:156-62

Hasannia E, Mohammadzadeh F, Tavakolizadeh M, Davoudian N, Bay M. Assessment of the anxiety level and trust in information resources among iranian health-care workers during the pandemic of coronavirus disease 2019. Asian J Soc Health Behav 2021;4:163-8

Olashore AA, Akanni OO, Fela-Thomas AL, Khutsafalo K. The psychological impact of COVID-19 on health-care workers in African Countries: A systematic review. Asian J Soc Health Behav 2021;4:85-97

Sharma R, Bansal P, Chhabra M, Bansal C, Arora M. Severe acute respiratory syndrome coronavirus-2-associated perceived stress and anxiety among indian medical students: A cross-sectional study. Asian J Soc Health Behav 2021;4:98-104

Chung, G. K.-K., Strong, C., Chan, Y.-H., Chung, Y.-N., Chen, J.-S., Lin, Y.-H., Huang, R.-Y., Lin, C.-Y., Ko, N.-Y. (2022). Psychological distress and protective behaviors during the COVID-19 pandemic among different populations: Hong Kong general population, Taiwan healthcare workers, and Taiwan outpatients. Frontiers in Medicine, 9, 800962. 

Lu, M.-Y., Ahorsu, D. K., Kukreti, S., Strong, C., Lin, Y.-H., Kuo, Y.-J., Chen, Y.-P., Lin, C.-Y., Chen, P.-L., Ko, N.-Y., Ko, W.-C. (2021). The prevalence of posttraumatic stress disorder symptoms, sleep problems, and psychological distress among COVID-19 frontline healthcare workers in Taiwan. Frontiers in Psychiatry, 12, 705657. 

Alimoradi, Z., Broström, A., Tsang, H. W. H., Griffiths, M. D., Haghayegh, S., Ohayon, M. M., Lin, C.-Y., Pakpour, A. H. (2021). Sleep problems during COVID-19 pandemic and its’ association to psychological distress: A systematic review and meta-analysis. EClinicalMedicine, 36, 100916. 

Author Response

Rebuttal letter

Anxiety, Depression and Posttraumatic Stress Disorder in

Physicians compared to Nurses during the Covid-19 pandemic

We thank the Referee for the valuables comments.

Ref. No. 2

 The submission has the following significant flaws.

Q1. The authors claimed that they recruited 700 participants; however, 95 physicians plus 675 nurses exceed 700.

R1. The data collection procedure was random, so only who wanted to voluntary answer to questionnaire was included in this study.

Q2. The entire manuscript is hard to follow as the authors use long paragraphs throughout the manuscript (e.g., only one paragraph in Introduction) 

R2. All the sections of the manuscript were revised according to the reviewer’s suggestions.

Q3. The authors did not control important confounders for the physician-nurse comparisons. They only used independent t-tests to examine the differences in the outcome measures. 

R3. The authors aimed to only photograph both anxiety and depression conditions among physicians and nurses. The enrollment procedure was voluntary, so any potential confounder variables were taken into consideration.

Q4. There are many unclear terms (e.g., STAY-1 and STAY-2; BDI). I know that the authors used Beck Depression Inventory; however, they did not introduce it. Instead, they said "Depression Inventory". Also, it is weird that they abbreviated State-Trait Anxiety Inventory into STAY instead of STAI. Moreover, they did not define what is STAY-1 and STAY-2.

R4. All the abbreviations were better explained.

Q5. There are some unclear sentences. For example, "The relationship between family caregiving and the mental health of emerging young adult caregivers [14–16]."

R5. All the manuscript was revised.

Q6. The authors mentioned "An observational, cross sectional, multicentric study, was carried out from March 2017 to December 2019 by administering an on line questionnaire in all Italian physicians and nurses who were employed during the Covid-19 pandemic." How can this happened? The pandemic begun from end of 2019!!

R6. Lines 93-94:from March 2020 to June 2020 by administering an on line questionnaire in all Italian physicians and nurses”.

Q7. There are quite a lot of typos (e.g., In the first art of the questionnaire; this should be In the first part of the questionnaire).

R7. All typos were adjusted.

Q8. Quite a lot of important references were not cited. Below are some examples.

R8. References suggested were also mentioned in our manuscript.

Changes and modifications have been marked in red color in the text to speed up reviewing.

I feel that the revised version of our manuscript, now, is ready for acceptance.

Sincerely yours,

Elsa Vitale

Bari, 13.05.2022

Reviewer 3 Report

The study benefits from large samples and very interesting scales that make it potentially a great study, however there are some major issues, here I tried to list them:

Abstract:

The background needs to be updated. The first two lines are not relevant.

Conclusions: remove the healthcare worker. The study populations were physicians and nurses; the conclusion is hard to read; please re-write it.

Study design: March 2017-December 2019

Line 122-127, please define items no. 5, 7, 8 , … or report the questionnaire as an appendix.

Method: The method section is so confusing; the survey was collected between March 2017 and December 2019; an important question is which part of these data was collected during the Pandemic; please explain that. If there is enough sample, the authors may compare the physiological disorder in physicians and nurses before and after the Pandemic.

Results: I suggest computing % of physicians and RN independently, for example, 43/95=45.2% and 529/675=78.3%, and then comparing the two-population using chi-sq or t-test.

I also suggest reporting the scales for all depressive scale metrics in a table or report as a boxplot to compare the PHY and RN.

I also suggest running several sets of regression using the scales as the dependent variable, RN (REF=PHY) as the primary independent variable, and control for all other socio-demographic or individual-level factors.

Discussion. Some reported studies measured depression in the general population, not in the HC workforce; please remove them.

Conclusion. The conclusion needs to be re-write; the current conclusion is not relevant to the study findings.

Author Response

Rebuttal letter

Anxiety, Depression and Posttraumatic Stress Disorder in

Physicians compared to Nurses during the Covid-19 pandemic

We thank the Referee for the valuables comments.

Ref. No. 3

The study benefits from large samples and very interesting scales that make it potentially a great study, however there are some major issues, here I tried to list them:

Q1. Abstract:

The background needs to be updated. The first two lines are not relevant.

Conclusions: remove the healthcare worker. The study populations were physicians and nurses; the conclusion is hard to read; please re-write it.

R1. The abstract section was revised according to the Reviewer’s suggestions.

Q2. Study design: March 2017-December 2019

Line 122-127, please define items no. 5, 7, 8 , … or report the questionnaire as an appendix.

R2. Items no….. were removed and authors decided to report only references for each of the questionnaire mentioned and administered.

Q3. Method: The method section is so confusing; the survey was collected between March 2017 and December 2019; an important question is which part of these data was collected during the Pandemic; please explain that. If there is enough sample, the authors may compare the physiological disorder in physicians and nurses before and after the Pandemic.

R3: Lines 93-94:from March 2020 to June 2020 by administering an on line questionnaire in all Italian physicians and nurses”.

Q4.: Results: I suggest computing % of physicians and RN independently, for example, 43/95=45.2% and 529/675=78.3%, and then comparing the two-population using chi-sq or t-test.

I also suggest reporting the scales for all depressive scale metrics in a table or report as a box plot to compare the PHY and RN.

I also suggest running several sets of regression using the scales as the dependent variable, RN (REF=PHY) as the primary independent variable, and control for all other socio-demographic or individual-level factors.

R4. As regards statistical analysis only chi square and t-test were performed.

Q5. Discussion. Some reported studies measured depression in the general population, not in the HC workforce; please remove them.

R5. The Discussion section was re-written according the Reviewer’s suggestions.

Q6. Conclusion. The conclusion needs to be re-write; the current conclusion is not relevant to the study findings.

 R6. Conclusions were re-written.

Changes and modifications have been marked in red color in the text to speed up reviewing.

I feel that the revised version of our manuscript, now, is ready for acceptance.

Sincerely yours,

Elsa Vitale

Bari, 13.05.2022

Reviewer 4 Report

  1. Methods: The aim of the present study was to analyze any differences existed between Italian physicians and nurses in anxiety, depression and PTSD levels during the Covid-19 pandemic. -please write the method used not the aim in this section
  2. It was difficult to understand and frame what was happening, 44 this new condition has generated isolation which has meant "putting on hold" an entire 45 organized life in this emergency.-please reframe
  3. from March 2017 to 83 December 2019- COVID-19 started in 2019, please explain the rationale of study starting from 2017.
  4. Deaths among health 183 workers are about 5 times higher than the average for the general population.-please cite
  5. The discussion is poorly cited and the results from 7 articles (21-27) have been discussed. These results can be presented in a table format. There are various published articles related to psychological/neurological sequalae of COVID-19, please discuss in detail.
  6. Please check for the English language as many incomplete sentences are there in the manuscript.
  7. Please check for yellow highlights as few examples

Author Response

Rebuttal letter

Anxiety, Depression and Posttraumatic Stress Disorder in

Physicians compared to Nurses during the Covid-19 pandemic

We thank the Referee for the valuables comments.

Ref. No. 4

Q1. Methods: The aim of the present study was to analyze any differences existed between Italian physicians and nurses in anxiety, depression and PTSD levels during the Covid-19 pandemic. -please write the method used not the aim in this section

R1. The method was written.

Q2. It was difficult to understand and frame what was happening, 44 this new condition has generated isolation which has meant "putting on hold" an entire 45 organized life in this emergency. -please reframe

R2. 44-45 lines were re-phrased.

Q3.  From March 2017 to 83 December 2019- COVID-19 started in 2019, please explain the rationale of study starting from 2017.

R3. Lines 93-94:from March 2020 to June 2020 by administering an on line questionnaire in all Italian physicians and nurses”.

Q4. Deaths among health 183 workers are about 5 times higher than the average for the general population. -please cite.

R4. This indication was cited.

Q5. The discussion is poorly cited and the results from 7 articles (21-27) have been discussed. These results can be presented in a table format. There are various published articles related to psychological/neurological sequalae of COVID-19, please discuss in detail.

R5. Results were better discussed.

Q6. Please check for the English language as many incomplete sentences are there in the manuscript.

R6. All the manuscript was completely revised in English language.

Changes and modifications have been marked in red color in the text to speed up reviewing.

I feel that the revised version of our manuscript, now, is ready for acceptance.

Sincerely yours,

Elsa Vitale

Bari, 13.05.2022

Round 2

Reviewer 2 Report

The main concerns are still there.

The authors recruited 95 physicians and 675 nurses. The total number exceeds 700. Apparently, the authors have data quality problem.

The authors did not control confounders to examine the differences between physicians and nurses. The results are thus not trustable. 

Reviewer 3 Report

Authors responded all of my comments, thank you!

Reviewer 4 Report

Most of the comments have not been addressed. The English language should be checked by a native English speaker. 

Line 26, the pandemic should be The pandemic

Please include the inclusion and exclusion criteria 

Please discuss the confounding factors and include a section "limitation of the study" to state that the confounding factors (list all) may have affected the results.

Line 116-117: please describe, the line is not conveying any meaning to me.

Line 125- administered? the literal meaning of administered is to inject or give something to patient, here used will be a better word

line 145- thanks to the- what does that mean? Are authors thanking SPSS? if yes, please include in acknowledgement section.

line 193-202: should be cited- there is no citation

what is the meaning of In 55 studies, in 57 studies- please clearly describe or explain

Overall, the manuscript need major formatting and revision for the English language.
